# Excalibur: A new ensemble method based on an optimal combination of aggregation tests for rare-variant association testing for sequencing data

**Simon Boutry**[1,2], **Raphaël Helaers**[1], **Tom Lenaerts**[2,3,4], **Miikka Vikkula**[1,5]*

**1** Human Molecular Genetics, de Duve Institute, University of Louvain, Brussels, Belgium, **2** Interuniversity Institute of Bioinformatics in Brussels, Université Libre de Bruxelles-Vrije Universiteit Brussels, Brussels, Belgium, **3** Machine Learning Group, Université Libre de Bruxelles, Brussels, Belgium, **4** Artificial Intelligence laboratory, Vrije Universiteit Brussel, Brussels, Belgium, **5** WELBIO department, WEL Research Institute, Wavre, Belgium

* miikka.vikkula@uclouvain.be

**Data Availability Statement:** The code, raw data and intermediate results to reproduce the preliminary test selection and the code, input data

## Abstract

The development of high-throughput next-generation sequencing technologies and large-scale genetic association studies produced numerous advances in the biostatistics field. Various aggregation tests, i.e. statistical methods that analyze associations of a trait with multiple markers within a genomic region, have produced a variety of novel discoveries. Notwithstanding their usefulness, there is no single test that fits all needs, each suffering from specific drawbacks. Selecting the right aggregation test, while considering an unknown underlying genetic model of the disease, remains an important challenge. Here we propose a new ensemble method, called Excalibur, based on an optimal combination of 36 aggregation tests created after an in-depth study of the limitations of each test and their impact on the quality of result. Our findings demonstrate the ability of our method to control type I error and illustrate that it offers the best average power across all scenarios. The proposed method allows for novel advances in Whole Exome/Genome sequencing association studies, able to handle a wide range of association models, providing researchers with an optimal aggregation analysis for the genetic regions of interest.

## Author summary

An increasing number of diseases previously thought to be caused by a mutation in a single gene are now being considered as involving several variants in a small number of genes (i.e. "oligogenic"). There is a limited number of dedicated bioinformatic tools to study such oligogenic causes of diseases. These include so called aggregation tests. Yet, an important challenge is to select the right aggregation test among the various ones that have been developed, as each suffers from different limitations. We have computationally compared 59 aggregation methods to explore their limitations. We found that combining 36 of them results in a more robust method, which we baptized "Excalibur". It can handle

from COSI and raw results from simulations pipeline to reproduce the numerical experiments simulations, are available on GitHub at https://github.com/dduv-ddit/excalibur_simulations or on figshare at 10.6084/m9.figshare.21780653.

**Funding:** This work was financially supported by the Fonds de la Recherche Scientifique - FNRS Grants T.0026.14 & T.0247.19, the Fund Generet managed by the King Baudouin Foundation (Grant 2018-J1810250-211305), and by la Région wallonne dans le cadre du financement de l'axe stratégique FRFS-WELBIO (WELBIO-CR-2019C-06) for WES sequencing of numerous human samples (all to MV). Simon Boutry was financially supported by fellowships from F.R.I.A. (Fonds pour la formation à la recherche dans l'industrie et dans l'agriculture), and Patrimoine UCL. The funders had no role in study design, data collection and analysis, decision to publish, or preparation of the manuscript.

**Competing interests:** The authors have declared that no competing interests exist.

a wider range of hypotheses and case-control studies than any of the single methods, while reducing the number of false positive results. Excalibur also provides a comprehensive elucidation of the underlying genetic architecture pertaining to each genomic region under investigation. Thus, it provides a user-friendly, and statistically sound platform to study oligogenic inheritance with the increasing amount of available genetic data.

## Introduction

Over the last decade, high-throughput next-generation sequencing (NGS) technologies have led to a significant increase in the quantity of genetic data available for statistical genetic analyses. Cost-effective whole genome (WGS) and whole exome (WES) sequencing have enabled large-scale genetic association studies (*e.g.* genome wide association studies (GWAS)), linking many common variants to complex traits: 4300 papers have reported 4500 GWASs and over 55 000 unique loci for nearly 5000 diseases and traits [1, 2].

Compared to common variants, rare genetic variants are more likely to be functional [3] and can more easily lead to novel biological and clinical insights [4]. However, methods for statistical analysis of rare variants are limited [5], calling for better genetic association analysis tools and functional experiments to obtain a more complete understanding of disease mechanisms [4]. To meet this goal, different new statistical methods, called aggregation tests, which analyze the association of a trait with multiple markers within a genomic region, have been proposed. Their strategy is to summarize the multiple genetic markers into a single "burden" score [6, 7] and analyze association of the trait with this score [8]. Given their potential to increase power to detect rare variant effects, region-based analysis has become the standard approach for analyzing rare variants in sequencing studies [5].

Many extensions and variations on the "burden approach" exist [5–7, 9–68]. They have been classified into different aggregation test categories (*e.g.* Burden, Variance-component, Omnibus, Data-driven and replication-based approaches) [5, 69, 70]. Each category and related methods have advantages and disadvantages [4, 8, 29, 39, 52, 70–74]. For example, Burden burden tests have diminished power when the effects of the genetic markers on the trait are in opposing directions, or when only small fractions of the variants influence the trait [69, 70, 75, 76].

Recognizing the inherent limitations of burden-based methods, Variance-component tests, such as the Sequence kernel Association test (SKAT), which builds upon the kernel machine regression framework to test rare variant associations, have been proposed [8, 13, 17, 20, 21, 24, 34, 35, 41, 42, 46, 50]. These Mixed Effect models investigate the distribution of rare variants between cases and controls, resolving the issue associated with variants of opposite effects [4, 8]. However, this approach loses power when the data has a large proportion of effects in the same direction [42, 57, 74].

To combine the strength of burden and variance-component tests, Omnibus tests were proposed, including the SKAT-Optimal test (SKAT-O) [41, 42], for situations in which both deleterious and protective variants are present within the same gene. Choosing the right statistical test given the nature of the data and the underlying genetic model of the studied disease (generally unknown) represents a crucial challenge for researcher aiming to apply aggregation tests. Sample size is generally also still an issue, because of sequencing costs and difficulties to have access to large amounts of patient data (especially for rare diseases). The performance of aggregation tests in small association studies is most of the time conservative [4, 8, 24, 41, 69].

Here, we investigate the limitations of current tests, analyzing the results obtained for different collections of simulated data. We assess type I error and empirical power of a large collection (n = 59) of established methods [5–7, 15, 20, 22, 25, 31, 35–37, 41, 42, 44–57, 59–62, 64, 66–68, 77–79]. The simulation method uses the backward coalescent model provided in the COSI 2 program [80, 81]. We investigated the behavior of each method along seven dimensions: [1] the proportion of cases and controls, [2] the cohort size, [3] the percentage of protective variants, [4] the percentage of causal variants, [5] the kind of variants (*i.e.* only rare variants, combining rare and common variants), [6] the causal minor allele frequency (MAF) cutoff, and [7] the size of the genetic region. We refer to the seven dimensions as limitations or scenarios. Moreover, we introduce and test four ensemble methods that each aim to overcome the limitations that most of the isolated aggregation tests are facing. We benchmarked these against the state-of-the-art using the same conditions [5–7, 12, 15, 20, 22, 25, 31, 35–37, 41, 42, 44–57, 59–62, 64, 66–68, 77, 78]. Our analysis provides a novel optimal ensemble method, which we refer to as Excalibur, which is able to control type I error and presents an increased average power across all performed simulations. Excalibur overcomes most limitations of classic aggregation tests, ensuring that researchers have easier access to high-quality results while being able to investigate a wider range of diseases and methodological assumptions. Moreover, Excalibur can be used to indicate which model and statistical test might be more suited for a particular set of data and genetic region.

## Material and methods

We consider $n$ individuals, split into two groups ($n_{cases}$ cases and $n_{controls}$ controls), sequenced for a genetic region with $m$ variant sites observed. For the i[th] subject, $y_i$ denotes the phenotype variable. Here we assume, without loss of generality, that the phenotypes are dichotomous (*e.g.* $y = 0/1$ for control and case respectively). Covariates can include gender, age or top principal components of genetic variation for controlling population stratification and are represented for individual $i$ by variable $X_i = (X_{i1}, X_{i2},..,X_{ip})$. The genotype of the $m$ variants within the genetic region for subject $i$ is denoted by $G_i = (G_{i1}, G_{i2},..,G_{im})$, with $G_{ij} = 0, 1$ *or* $2$ represents the number of copies of the minor allele. To link the sequenced variants in a genetic region to the phenotype, we consider the logistic model:

$$logit(y_i = 1) = \beta_0 + \boldsymbol{\alpha} \, X_i + \boldsymbol{\beta} \, G_i$$

where $\beta_0$ is an intercept term, $\boldsymbol{\alpha} = [\alpha_1,..,\alpha_p]$ is the vector of regression coefficients for the $p$ covariates, and $\boldsymbol{\beta} = [\beta_1,..,\beta_m]$ is the vector of regression coefficients for the $m$ observed variants in the genetic region.

The main hypothesis in this paper, as suggested by other authors [6, 41, 59], is that combining several aggregation tests in an ensemble method will increase power while still controlling for type I error.

## Preliminary test selection

We made a selection of different aggregation tests, all freely available in R packages, and able to handle dichotomous traits [5–7, 12, 15, 20, 22, 25, 31, 35–37, 41, 42, 44–57, 59–62, 64, 66–68, 77, 78]. After literature review, we identified 10 R packages implementing 154 aggregation tests (see S1 Table). For the sake of scalability, we performed a preliminary selection, divided into 4 steps, to select the most computationally efficient methods. The first step uses the data publicly available within the SKAT package [35, 41, 42, 46, 54] to filter out the tests that did not work or were computationally too demanding. The SKAT data consists of a matrix $Z$, i.e., a numeric genotype matrix of 2000 individuals and 67 SNPs. Each row and each column

represent a different individual and a different SNP, respectively. We use $y. b$, a numeric vector of binary phenotypes. Based on these data, we performed 5 analyses with different cohort sizes (100, 500, 1000, 1500, and 2000), ran each test, and retrieved the p-values and computational time for each. We removed tests with mean computational time above 10 seconds (step 1.A: 19 tests removed), maximum computational time above 10 seconds (step 1.B: 22 tests removed) and tests that did not work (step 1.C: 6 tests removed). The size of the genotype matrix did not correlate with computational time explosion for DoEstRare [22]. We ran a separate analysis to investigate that test (step 1.D) resulting in removing this test as a potential candidate due to scalability concerns.

The second step consisted in computing the difference and redundancy (with redundancy defined as the number of times two tests had the same p-value for the same analysis) for the 106 remaining tests. Tests being redundant with others were removed based on the maximum evolution of computational time to keep only the most computationally efficient ones. This was computed as $\text{Max\_evol} = \frac{Max(time) - Min(time)}{Min(time)}$

In total 18 tests were removed in this second step.

The third analytical step was based on the same data as in step 1 and 2, but extending the genotype matrix $Z$ and phenotype vector $y. b$ to 4000 and 8000 individuals (conserving a balanced cohort). For the new individuals, SNPs in the genotype matrix were randomly generated. We ran the remaining 88 tests on these two datasets to assess the computational time, removing again tests having both a maximum time above 10 sec and evolution above 10 (step 3: 11 tests removed).

Through these three steps, computationally prohibitive aggregation tests were removed. To ensure that there were no redundant tests left, we performed a final step (step 4 including 77 tests). We ran a power simulation using our state-of-the-art simulation framework (see Table 1 and S1 Fig). Based on 3000 repeats, this preliminary power simulation provided sufficient data to compute the differences and redundancies between each of them. We used the same criteria from step 2 to decide whether to keep or remove a test. We obtained 59 computationally scalable and non-redundant tests that together constituted the potential candidates for the ensemble approach.

Detailed values of each parameter for the 9 type I error experiments grouped into 4 scenarios and the 18 empirical power experiments grouped into 7 scenarios, and the power analysis performed for preliminary test selection.

## Overcoming limitations of aggregation tests with our ensemble methods

Based on the 59 aggregation tests, an ensemble method was created with the ability to overcome the main (classical) limitations of individual methods. Our two criteria to evaluate these methods were the ability to control type I error and achieving the best average power across all simulations. We investigated several possible ensemble methods, and present four of them. The first method, called Excalibur_baseline, is a baseline combination of all 59 state-of-the-art methods (S2 Table). The second method, called GoodTypeI_Excalibur, was constructed as follows:

1. From **Set A** = {59 state-of-the-art methods}, we selected the ones with a proportion of inflated type I error equal to zero (*i.e*. 31 methods), which will be referred to as **Set B**

2. From **Set B**, across all empirical power simulations,

   a. We computed a **score** defined as the number of times a test was the only one having a significant p-value (out of the tests in **Set B**)

   b. All tests that had a **score** equal to zero (*e.g*. CAST fisher and chisq) were removed, producing a new **Set C**

**Table 1. Description of experiments.**

| | Type I error experiment ID | | | | | | | | | Empirical power experiment ID | | | | | | | | | | | | | | | | | | Power for preliminary test selection |
|---|---|---|---|---|---|---|---|---|---|---|---|---|---|---|---|---|---|---|---|---|---|---|---|---|---|---|---|---|
| Scenario | 1 | 2 | 3 | 4 | 5 | 6 | 7 | 8 | 9 | 1 | 2 | 3 | 4 | 5 | 6 | 7 | 8 | 9 | 10 | 11 | 12 | 13 | 14 | 15 | 16 | 17 | 18 | |
| 1. Proportion cases/controls | 0.5 | | | | | 0.2 | 0.8 | 0.5 | | 0.5 | | | | | | | | | | | | | 0.2 | 0.8 | 0.5 | | | 0.5 |
| 2. Cohort size | 100 | 5000 | 500 | 1000 | | 500 | | 1000 | 5000 | 1000 | | | | | 100 | 5000 | 500 | 500 | | 1000 | | | | | 500 | | 1000 | 500 |
| 3. Percentage protective | 5 | | | | | | | | | 5 | | | | | 0 | | | | 5 | | 10 | 20 | | | 5 | | | 5 |
| 4. Percentage causal | 20 | 1 | 5 | 10 | 20 | | | | | 20 | 1 | 5 | 10 | 20 | | | 5 | | | | | | 20 | | | | 5 | 20 |
| 5. Rare / Common | only rare | | | | both | only rare | | | | only rare | | | | | | | | | both | | | | only rare | | | | | only rare |
| 6. Causal MAF cutoff | 0.01 | | | | | | | | | 0.03 | | | | | | | | | | | | | | | | | | 0.03 |
| 7. Region size | 3000 | | | | | | | 1000 | 5000 | 3000 | | | | | | | | | | | | | | 5000 | | | 3000 | 3000 |

3. From **Set C**, we defined the reliability of a test as the sum of significant and non-significant p-values divided by the number of replicates. Methods with median reliability above 0.9 were kept, thus removing cmat and VT, for instance.

The GoodTypeI_Excalibur ensemble method thus consists of 24 tests (S2 Table).

A third ensemble, which we called Excalibur, was constructed by expanding the second ensemble method, using the following selection steps:

1. We started with tests present in GoodTypeI_Excalibur as **candidates**, other tests = **Set A**

2. From **Set A**, we selected test(s) with the minimum proportion of inflated type I error, **Set B**

3. From **Set B**, across all empirical power simulations,

   a. computed a **score** defined as the number of times a test was the only one having a significant p-value (out of the tests in **Set B**)

   b. From **Set B**, selected test(s) with maximum **score** and added it/them to **candidates**, and updated **Set A** by removing **candidates** from **Set A**

4. Recomputed type I error for all simulations for the updated **candidates**

   a. If the proportion of inflated type I error equaled to zero, we restarted the procedure at step 2

   b. Otherwise, we stopped the procedure, and removed the last **candidates** from **Set A**. This **Set A** constituted the new ensemble method to be tested

As the algorithm successfully looped 12 times, Excalibur ensemble method consists of 36 tests (S2 Table).

The $4^{st}$ ensemble method, is another attempt to expand the second ensemble (*i.e.* GoodTypeI_Excalibur) in an orthogonal manner:

1. We started with tests present in GoodTypeI_Excalibur as **candidates**, other tests = **Set A**

2. From **Set A**, across all empirical power simulations,

   a. computed a **score** defined as the number of times a test was the only one having a significant p-value (out of the tests in **Set A**).

   b. Computed their **rank**, i.e. in order of decreasing score.

   c. Computed a **rank type I,** ranging from the smallest proportion of inflated type I error to the biggest.

   d. Established a **combined rank** as the sum of **rank** and **rank type I**.

3. From **Set A**, selected test(s) with minimum **combined rank** and added it (them) to **candidates**, and updated **Set A** by removing **candidates** from **Set A**

4. Same as Step 4 (see above)

This implementation successfully looped 4 times, generating the new ensemble method called V2_Excalibur, based on 28 tests (see S2 Table).

Additional ensemble creation methods were explored but did not result in better ensembles than GoddTypeI_Exaclibur and therefore are not presented here. P-values for our ensemble methods are computed as the minimum p-value from the set of tests included in the ensemble method after multiple testing correction using Bonferroni.

## Design of numerical experiments and simulations

In this section we describe the design of numerical experiments and simulations. To construct and validate our Excalibur ensemble method in terms of protecting type I error and to assess its power compared to the 59 tests, we used our simulation pipeline (see S1 Fig) to carry out simulation studies under a wide range of experiments (see Table 1). For each simulation, we determined sequence genotypes using COSI 2 [80, 81]. We simulated 10,000 COSI haplotypes, each resembling a 1 million base pair region based on COSI's coalescent model that mimics the local recombination rate based on the linkage disequilibrium pattern and the population history for Europeans. For each simulation, we set disease prevalence $\beta_0$ to 1%, as previously done in [22, 41, 42, 46, 59, 82, 83], and $\boldsymbol{\alpha} = (0.05, 0.01, 0.001)$.

## Type I error simulations

To investigate whether our ensemble method and the aggregation tests under study preserved the desired type I error rate under various scenarios, we ran 9 experiments (see Table 1). We tested 4 input parameters potentially having an impact on type I error:

1. Case $n_{cases}$ and control $n_{controls}$ proportion within the cohort, using proportions of cases being set to 0.2, 0.5 or 0.8

2. Cohort size [5, 36, 41, 42, 46, 60–62, 83–86]: varying the total number of individuals ($n$) from 100, 500, 1000 to 5000.

3. Including or excluding common variants (i.e., MAF > causal MAF cutoff) [59, 60, 62]: taking into account only rare variants (*MAF<causal MAF cutoff*) or including also common variants (*MAF>causal MAF cutoff*)

4. Size of the regions to be analyzed [41, 42, 49, 55, 83]: testing with 1K, 3K and 5K base pair regions

We performed 10000 simulations for each experiment. To evaluate type I error, we estimated the empirical type I error rate as the proportion of p-values less than α, as proposed in [22, 41, 42, 46, 83].

For each simulation, we randomly selected regions of given size while ensuring a minimum number of variants ($m \geq 2$), minimum number of causal variants ($m_c$), and different regions for each simulation performed within the same experiment [22, 41, 42, 46, 59].

To evaluate type I error, datasets under the null model and dichotomous phenotypes (given the desired cohort size, and proportion of cases ($n_{cases}$) and controls ($n_{controls}$)) were generated from the null logistic regression model

$$y_i = logit^{-1}(\beta_0 + 0.5\,X_{i1} + 0.5\,X_{i2})\forall i \in 1, .., n$$

where $X_{i1}$ was a continuous covariate from N(0,1), and $X_{i2}$ was a binary covariate from Bernoulli(0.5), as proposed [22, 41, 42, 46, 54, 56, 59, 60, 62, 82, 83].

## Empirical power simulations

To investigate the empirical power of our ensemble method and the aggregation tests under various scenarios, we ran 18 experiments (see Table 1). We tested seven input parameters potentially having an impact on the empirical power:

1. Case $n_{cases}$ and control $n_{controls}$ proportion within the cohort, using proportions of cases being set to 0.2, 0.5 or 0.8

2. Cohort size [5, 36, 41, 42, 46, 60–62, 83–86]: varying the number of individuals ($n$) from 100, 500, 1000 to 5000.

3. Percentage of protective variants [24, 36, 41, 42, 46, 49, 54, 59, 83]: percentage of causal variants with a protective effect set to 0%, 5%, 10% or 20%

4. Percentage of causal variants [22, 24, 41, 42, 46–49, 54, 58, 83, 87]: percentage of variants ($m$), with *MAF<causal MAF cutoff*, being assigned as causal, set to 1%, 5%, 10% or 20%

5. Including or excluding common variants [59, 60, 62]: taking into account only rare variants (*MAF<causal MAF cutoff*) or including also common variants (*MAF>causal MAF cutoff*)

6. Causal MAF cutoff [6, 24, 36, 41, 42, 46, 48, 54, 55, 59, 62, 83, 87]: threshold separating rare and common variants, set to 0.01 or 0.03

7. Size of the genetic region [41, 42, 49, 55, 83]: testing with 3K and 5K base pairs

We performed 1000 simulations per experiment. We estimated the empirical power as the proportion of p-values less than α, as proposed in [22, 41, 42, 46, 83]. For each simulation, we randomly selected regions of given size while ensuring a minimum number of variants ($m\geq 2$), minimum number of causal variants ($m_c$), and different regions for each simulation perform within the same experiment. We randomly assigned causal variants among the ones with *MAF<causal MAF cutoff*. Datasets under the alternative modeland dichotomous phenotypes (given the desired cohort size, proportions of cases ($n_{cases}$) and controls ($n_{controls}$)) were generated from the logistic regression model

$$y_i = logit^{-1}(\beta_0 + 0.5\,X_{i1} + 0.5\,X_{i2} + \sum_{j=1}^{m_c} \beta_j\,G_{ij})\forall i \in 1,..,n$$

where $X_{i1}$ was a continuous covariate from N(0,1), and $X_{i2}$ was a binary covariate from Bernoulli(0.5), as proposed [22, 41, 42, 46, 54, 56, 59, 60, 62, 82, 83]. $G_{ij}$ are the genotypes of the $i$ causal rare variants (randomly selected subset of the simulated rare variants). For all aggregation tests under study (and able to integrate a weighting scheme of the variant), equation $|\beta_j| = c|\frac{\log\,(MAF_j)}{2}|\forall j \in 1,..,\mathrm{m}_c$ was used to set $\beta$ as the effect size for causal variants and up weight rarer variants, as proposed [41, 42, 46, 83]. We scaled down c with larger percentage of causal variants (*e.g* c $= \frac{\ln\,(5)}{4}$, $c = \frac{\ln\,(7)}{4}$ and $c = \frac{\ln\,(13)}{4}$ when the percentage of causal variants was 20%, 10% or 5% respectively, as proposed [41, 42, 46, 82].

## Results

### Type I error

We assessed type I error for all the tests over nine experiments, each based on 10 000 replications, accounting for 90 000 simulations in total (see Table 1). For each simulation and each test, type I error was evaluated given three $\alpha$ levels (0.05, 0.01, 0.001). The detailed results are in S3 Table. Several terms to analyze type I error results need to be noted:

• *Significant p-value* was a p-value below or equal to the $\alpha$ level under consideration, otherwise the p-value was considered not significant.

• *NA* if a test failed to return a p-value

• *Type I error* was equal to the number of significant p-values divided by the number of returned p-values (for a particular experiment and a given $\alpha$ level). Returned p-values equal the number of replicates (10 000) times reliability.

Type I error is thus not defined as the number of significant p-values divided by 10 000 (number of replicates) as this would result in an overestimation. NA is used to establish the reliability (see section material and methods) of the test for a particular experiment. The goal was to find the aggregation test that can controls type I error across all experiments and $\alpha$ levels (Fig 1 and supporting information in S3 and S4 Tables). In total, 27 type I error results

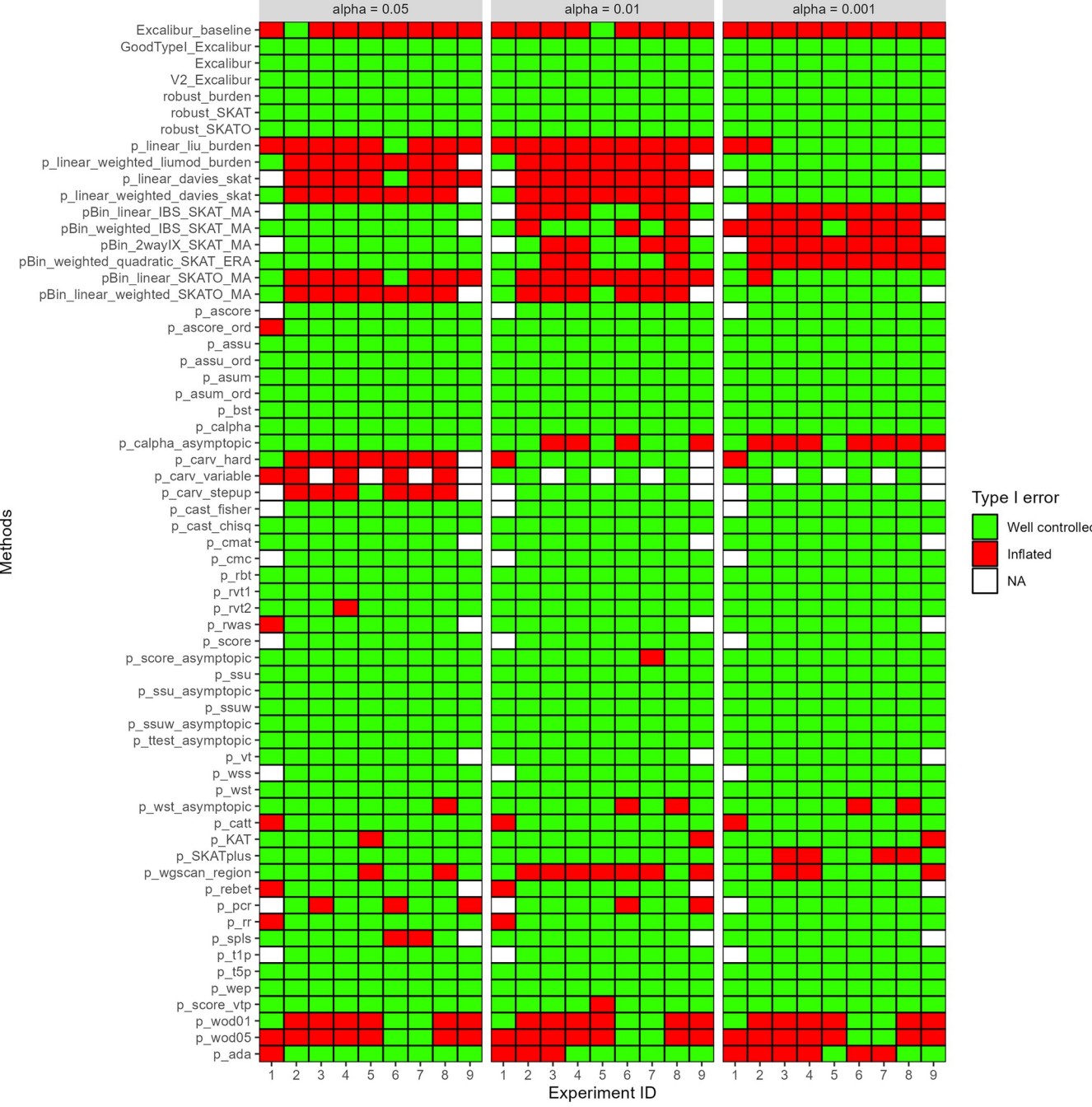

**Fig 1. Summary of type I error for nine experiments.** X-axis is the experiment ID (see description in Table 1) evaluated at three $\alpha$ levels (*e.g.* 0.05, 0.01 and 0.001) for our 4 ensemble methods and 59 state-of-the-art aggregation methods (see Y-axis). Green: type I error below the $\alpha$ level threshold; therefore labelled as well controlled. Red: type I error above the $\alpha$ level threshold; therefore labelled as inflated. NA: test failed to return p-values.

were generated per test (*i.e.* 9 experiments, each evaluated at three different alpha levels). We defined:

- *Proportion good* as the number of times a test managed to control type I error divided by 27, i.e. the number of type I error results generated per test. "Good" or "well-controlled" were used as synonyms

- *Proportion inflated* is the proportion, out of the 27 results, in which a test had an inflated type I error

- *Proportion conservative* is the proportion, out of the 27 results, in which a test had a conservative type I error

- *Proportion of NA* is defined as the proportion a test fails to return a p-value

- The minimum, maximum and median reliability across all results are shown as well.

For each $\alpha$ level, we defined a confidence interval assuming that the type I error followed a binomial distribution with parameter 10000 (*i.e.*, number of replicates per experiment) and $\alpha$ level. We defined an $\alpha$ level threshold as the upper limit of this interval, namely, 0.054, 0.0121 and 0.0018 for $\alpha$ levels 0.05, 0.01 and 0.001, respectively (see red cases in Fig 1 and upper blue lines in S2 and S3 Figs). Therefore, a test had an inflated type I error for a particular experiment if its type I error was above that $\alpha$ level threshold. We defined an $\alpha$ level threshold as the lower limit of this interval, namely, 0.046, 0.0079 and 0.0002 for $\alpha$ levels 0.05, 0.01 and 0.001, respectively (see bottom blue lines in S2 Fig and S3). Therefore, a test had a conservative type I error for a particular experiment if its type I error was below that $\alpha$ level threshold.

The results in Fig 1 show that 20 state-of-the-art methods and 3 versions of our ensemble methods could control type I error across all experiments and $\alpha$-values. In addition, these tests exhibited a median reliability above 99%. An additional eight methods did not have inflated type I error but failed to retrieve a p-value in one experiment. Among them, five tests (*i.e.* p_ascore, p_cast_fisher, p_cmc, p_score, p_wss and p_t1p) had a reliability of 99% and can be considered to control well type I error, except in small cohorts. The other two methods (*i.e.* p_cmat and p_vt) suffered from a 2% reliability median. In total, 22 tests did not work each time (see Fig 1) and were unable to return a p-value for some simulations. This discovery we encountered during our investigation is both intriguing and merits further inquiry. However, we acknowledge that such an exploration lies outside the scope of the present manuscript. For example, type I error experiment ID n˚9 presents up to 12 tests that were not successful. Because of our definition of type I error, tests that had a poor reliability had an increased type I error, in some cases leading even to a type I error of 1 (*e.g.* p_carv_variable and p_carv_hard, which both present the lowest median reliability, S4 Table). These results reveal the inherent difficulty of choosing the right test given a particular analysis and that a naïve combination of all tests (named Excalibur_baseline) leads to an inflated type I error.

## Empirical power

We assessed empirical power for all tests over 18 experiments, following a standard procedure with each based on 1000 replications, producing 18,000 simulations in total. For each simulation and each test, the empirical power was evaluated at three $\alpha$ levels (*i.e.* 0.05, 0.01, 0.001). The detailed results are provided in S5 Table. We defined:

- *Significant p-value* as a p-value below or equal to the $\alpha$ level under consideration, otherwise the p-value was considered not significant.

- *NA* if a test failed to return a p-value

- *Power* is the number of significant p-values divided by 1000 (*i.e.* the number of replicates for a particular experiment)

Our goal was to find tests achieving the greatest power while controlling type I error across all experiments and $\alpha$ levels. We separated empirical power results:

- Good type I error (see S4 Fig): a test having the proportion of inflated type I error equal to zero (see S4 Table)

- Badly controlled type I error (see S5 Fig): tests having the proportion of inflated type I error above zero (see S4 Table). These tests might, in some experiments, have an inflated type I error.

We evaluated the power of each test for each experiment [18] given each $\alpha$ level [3] and giving 54 empirical power results per test. As shown in S5 Table, at $\alpha = 0.05$, all methods had a high standard deviation, except for methods that perform poorly on all experiments. This demonstrates the impact of the parameters on each experiment (see Table 1), and highlights the difficulty of choosing the right statistical test. Note that experiments with ID n˚6 and 10 exhibited the worst average power across all tests, with 0.109, while experiments with ID n˚5, 11 and 16 had the best average power across all tests, 0.475, at nominal $\alpha = 0.05$. This indicates a dramatic impact when including non-causal variants in the analysis, and additionally underlines the importance of an efficient variant filtering step (or variants selection methods) before running any aggregation test.

For each empirical power experiment, each test was ranked according to its power (S6 Table). Based on these rankings, we computed an average, best and worst ranking achieved by each test across all empirical power results (S7 Table). We found that the best average ranking method with good type I Error are our three other ensemble methods (*i.e.* Excalibur, V2_Excalibur, GoodTypeI_Excalibur, with average rank of 3.9, 4.4 and 5.6, respectively).

Comparing S4 and S7 Tables reveals the importance of performing a preliminary type I error analysis. For example, in the top 8 average ranked tests in S7 Table, p_wgscan_region, pBin_ _weighted_IBS_SKAT_MA, p_linear_weighted_davies_skat and pBin_linear_weighted_SKATO_MA exhibit a good type I error in only 18 out of 27 experiments at best, while the Excalibur_baseline method only achieves that in 1 experiment (Fig 1). One should thus be careful when comparing tests based on their empirical power without considering their type I error.

Without going into the details of the extensive literature [88–90] on the relationship between type I and type II errors (*type II error* = (1−*Power*)), increasing the power of a test generally leads to an increase of type I error. It is preferable to deal with a smaller power than an inflated type I error. In other terms, it is better to have a small number of reliable results, than numerous results with poor confidence, as aggregation tests aim to explore genomic data to guide researchers towards further in-vitro experiments. Fig 2 shows the proportion of experiments where type I error was well controlled along with the average power across all experiments for our four ensemble methods and the 59 state-of-the-art methods. Except for Excalibur_baseline, all three ensemble methods control type I error while improving the power, with Excalibur being the best. In summary, this analysis allowed us to establish a test performing on average with the best rank in power across all experiments and alpha levels, while controlling type I error in each experiment. In the 18 empirical power experiments, the best three methods are always our three ensemble methods, except for experiments ID n˚ 10 where robust_SKAT and robust_SKATO are better.

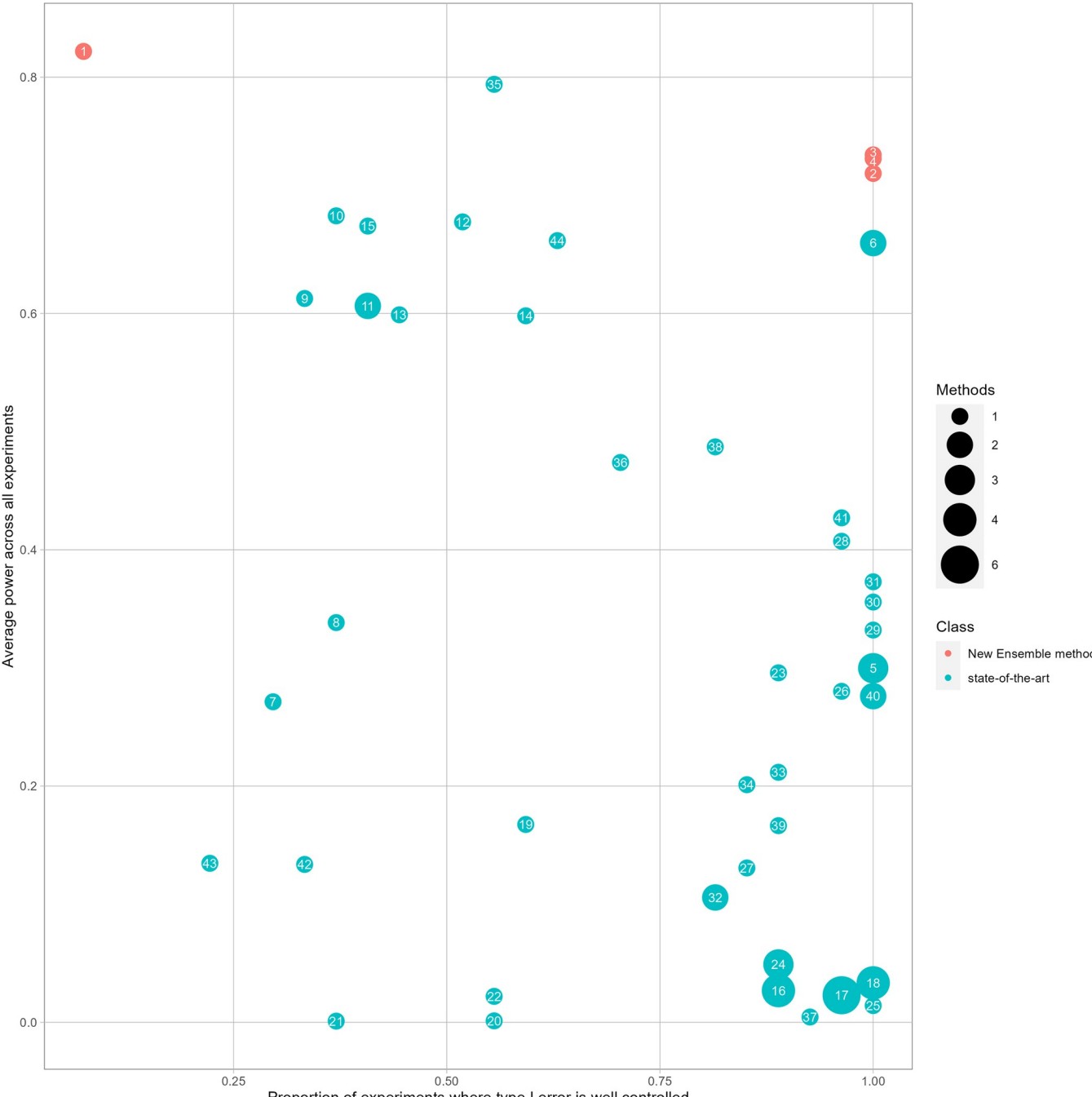

**Fig 2. Proportion of experiments where type I error is well controlled and the average power.** Proportion of experiments where type I error was well controlled (see x-axis) and the average power (see y-axis) computed across all experiments [18] and $\alpha$ levels [3] (54 results per tests) for our 4 ensemble methods (in red) and 59 state-of-the-art methods (in turquoise). To improve visual interpretability, some methods were grouped (see Methods for the number being grouped). Legend ID 1) Excalibur_baseline 2) GoodTypeI_Excalibur 3) Excalibur 4) V2_Excalibur 5) robust_burden, p_cast_chisq, p_rvt1 6) robust_SKAT, robust_SKATO 7) p_linear_liu_burden 8) p_linear_weighted_liumod_burden 9) p_linear_davies_skat 10) p_linear_weighted_davies_skat 11) pBin_linear_IBS_SKAT_MA, pBin_linear_SKATO_MA 12) pBin_weighted_IBS_SKAT_MA 13) pBin_2wayIX_SKAT_MA 14) pBin_weighted_quadratic_SKAT_ERA 15) pBin_linear_weighted_SKATO_MA 16) p_ascore, p_cmc, p_score, p_wss 17) p_ascore_ord, p_assu, p_asum_ord, p_calpha, p_ssuw, p_wst 18) p_assu_ord, p_asum, p_bst, p_ssu, 19) p_calpha_asymptopic 20) p_carv_hard 21) p_carv_variable 22) p_carv_stepup 23) p_cast_fisher 24) p_cmat, p_vt, p_catt 25) p_rbt 26) p_rvt2 27) p_rwas 28) p_score_asymptopic 29) p_ssu_asymptopic 30) p_ssuw_asymptopic 31) p_ttest_asymptopic 32) p_wst_asymptopic, p_rebet 33) p_KAT 34) p_SKATplus 35) p_wgscan_region 36) p_pcr 37) p_rr 38) p_spls 39) p_t1p 40) p_t5p, p_wep 41) p_score_vtp 42) p_wod01 43) p_wod05 44) p_ada.

In S6 Fig, we conducted a comprehensive analysis of 59 state-of-the-art methods using principal component analysis (PCA) based on 18,000 empirical power simulation results. Different implementations of the SKAT methods exhibit clustering patterns (*e.g.* pBin_weighted_quadratic_SKAT_ERA, pBin_2wayIX_SKAT_MA and pBin_linear_IBS_SKAT_MA). In order to assess the proportion of similarity among two tests, we compared the decision (p-value significant or not) evaluated at $\alpha$ = 0.05 for 18,000 empirical power simulation. S7 Fig shows the proportion of similarity above 0.5 of our 4 ensemble methods and 59 state-of-the-art tests. S8 Fig is similar to S7 Fig, while performing a hierarchical clustering and focusing on 36 state-of-the-art methods included in Excalibur. Most methods have a proportion of similarity bellow 0.5, with the highest similarity (*i.e.* 0.96) achieved by p_cast_chisq and p_rvt2.

## Exploring some limitations of aggregation tests

To investigate the impact of some of the limitations on the performance of the aggregation tests, exhaustive simulations covering four type I error scenarios and seven empirical power scenarios were performed. S8 Table shows the different experiments in order to compare the impact of a single parameter value on the behavior of the tests (type I error and empirical power). Because all other parameters were set equal in all experiments within a scenario (Table 1), parameter by parameter conclusions can be drawn. The detailed results are provided in S9 and S10 Tables. We defined:

- *Evolution* as the type of change (*e.g.* increase, decrease, no change, . . .) in type I error or empirical power when increasing a parameter value

- *Total evolution* as the sum of changes in type I error or empirical power when changing a parameter value (*e.g.* when increasing the proportion of cases within the cohort)

For each scenario, we evaluated our 4 ensemble methods and the 59 state-of-the-art tests at three $\alpha$ levels. For example, S9 and S10 Figs show type I errors and empirical power results at nominal level $\alpha$ = 0.05 for the scenario in which cohort size was increased. Data in S11 Table confirms that increasing a parameter value (in this case the cohort size) can have a heterogeneous effect on the test behavior (even within the same test). The total power evolution gave a global overview of the impact of changing a parameter value on a test behavior and led to the following observations (Fig 3, S12 Table and S11 Fig). Here bellow we only discuss tests with a proportion of inflated type I error equal to zero, giving each time the 5 best state-of-the-art methods (based on their power):

1. Increasing the proportion of cases within the cohort led to an increase in empirical power for the 4 ensemble methods and 26 of the state-of-the-art methods (up to 0.4 increase in power for p_cast_chisq), while decreasing the power of the other tests (up to -0.57 decrease in power for p_ssuw_asymptopic). We identified that robust_SKATO and robust_SKAT perform best regardless of the proportion of cases to be low (0.2) or high (0.8). For low number of cases, the next best methods were p_ssuw_asymptopic, p_ttest_asymptopic and p_ssu_asymptopic, while for high number of cases, the next best methods were p_cast_fisher, p_cast_chisq and p_wep.

2. One could expect that increasing cohort size should lead to an increase in power, but our results show that this does not hold for nine of the methods. For example, p_rbt suffered from a 0.056 decrease in power, while the average gain of power across all tests was 0.16. Note that the GoodTypeI_Excalibur power was increased up to 0.84 when increasing the cohort size. For a small cohort (100 individuals), we identified as the best methods p_ssu_asymptopic, p_rbt, p_score, robust_SKATO, and robust_SKAT. For a large cohort (5000

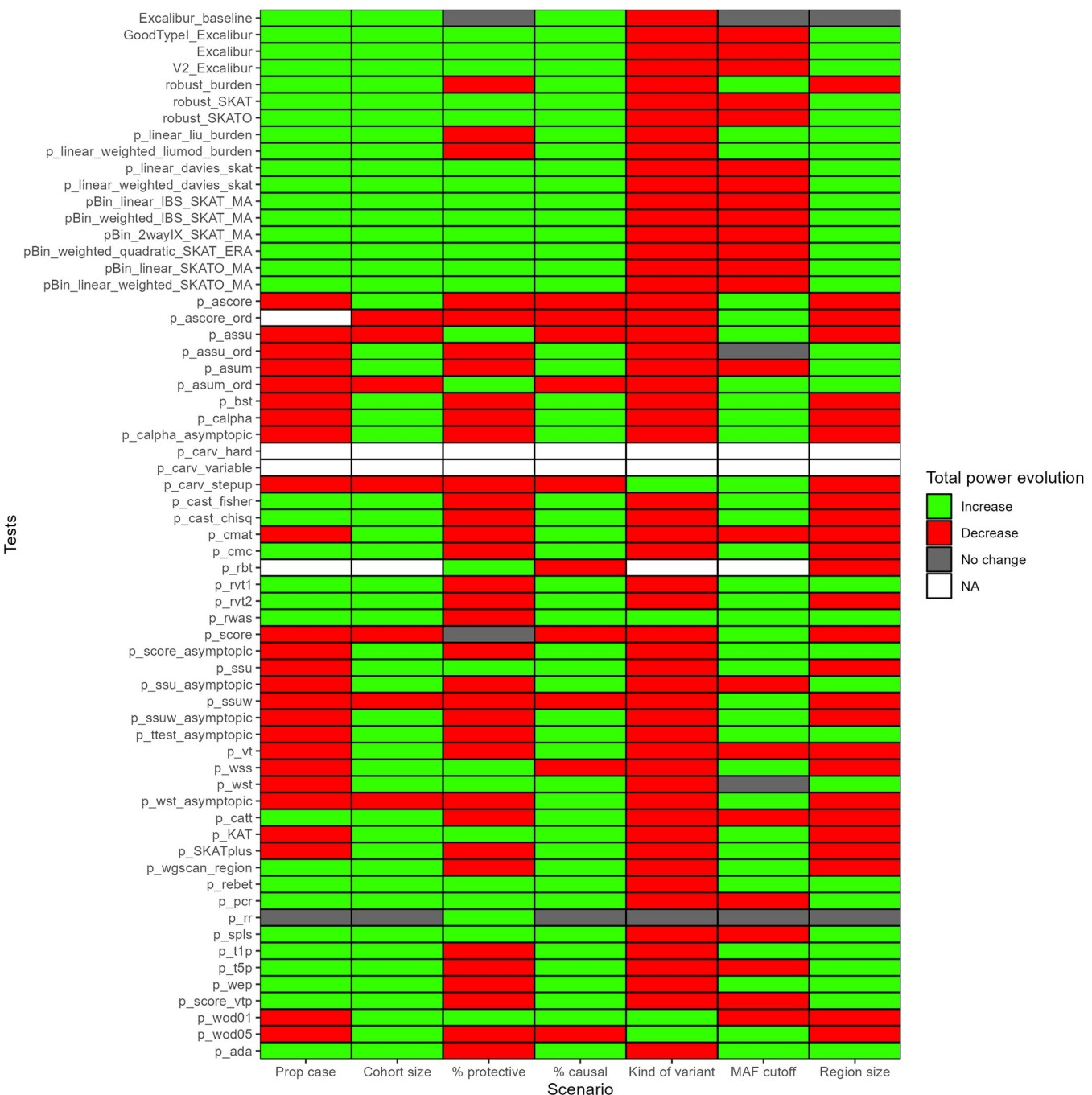

**Fig 3. Summary of power evolution across scenarios.** Total evolution of empirical power for seven scenarios (see X axis) for our 4 ensemble methods and 59 state-of-the-art methods (see Y axis) at nominal level α = 0.05 (data in S12 Table). Green: total power evolution above zero. Red: total power evolution below zero. Grey: total power evolution equal to zero. NA: no information for total power evolution. Prop case: evolution of power given the evolution of proportion of case in the cohort, based on empirical power ID n˚14, n˚11 and n˚15 (Tables 1 and S5). Cohort size: evolution of power given the evolution of cohort size, based on empirical power ID n˚6, n˚7, n˚8 and n˚9 (Tables 1 and S5). % protective: evolution of power given the evolution of proportion of protective variants, based on empirical power ID n˚11, n˚12 and n˚13 (Tables 1 and S5). % causal: evolution of power given the evolution of proportion of causal variants, based on empirical power ID n˚2, n˚3, n˚4 and n˚5 (Tables 1 and S5). Kind of variant: evolution of power given the inclusion of only rare variants versus rare and common variants, based on empirical power ID n˚18 and n˚10 (Tables 1 and S5). MAF cutoff: evolution of power given the evolution of causal MAF cutoff, based on empirical power ID n˚1 and n˚11 (Tables 1 and S5). Region size: evolution of power given the evolution of region size, based on empirical power ID n˚17 and n˚16 (Tables 1 and S5).

individuals), the best methods were robust_SKAT, robust_SKATO, p_ttest_asymptopic, p_ssuw_asymptopic, p_cast_fisher and p_cast_chisq.

3. In the presence of an increasing number of protective variants, the average total power evolution across methods remained equal, except for 8 methods (all belonging to the Burden test class), with the worst decrease in power of 0.15 for p_wep. There was nearly no gain of power attributed to protective variants (*i.e.* best total evolution of power was only 0.04 for robust_SKAT). Regardless of the proportion of protective variants, the best methods were robust_SKAT, robust_SKATO, p_cast_fisher, p_cast_chisq and p_rvt1.

4. On average, the most impactful parameter regarding the total evolution of power was the percentage of causal variants, ranging from a decrease of 0.01 for p_asum_ord to an increase of 0.74 for robust_SKATO, stressing the importance of a decent method for variant filtering and selection prior to running an aggregation test. With a small percentage of causal variants (1%), the best methods where robust_SKAT, robust_SKATO, p_ssuw_asymptopic, p_ttest_asymptopic, and p_ssu_asymptopic. On the other hand, with a high percentage of causal variants (20%), the best methods were robust_SKATO, robust_SKAT, p_cast_fisher, p_cast_chisq and p_rvt1

5. All methods lost power when introducing common variants, up to 0.5 for V2_Excalibur. There was no clear gain of power linked to the introduction of common variants. The best methods to handle only rare variant were robust_SKAT, robust_SKATO, p_ttest_asymptopic, p_ssuw_asymptopic and p_ssu_asymptopic. The best methods to handle a mixture of common and rare variants were robust_SKATO, robust_SKAT, p_wep, p_t5p and p_rvt1.

6. On average, relaxing the MAF cutoff from 0.01 to 0.03 led to very small increase in total evolution of power (0.04) and very limited decrease (worst decrease was attributed to p_ssu_asymptopic with -0.127), while both CAST methods (p_cast_fisher and p_cast_chisq) presented a total evolution of 0.32. When using a small cutoff (0.01) for the MAF, the best methods were robust_SKATO, robust_SKAT, p_wep, p_t5p and p_ssu_asymptopic.

7. The size of the genetic region was, on average the less impactful parameter (gain of 0.02), ranging from a limited loss of 0.097 (for p_ssuw_asymptopic) to a maximum total power evolution of 0.16 (for p_ssu_asymptopic). For regions of 3,000 base pairs, the best methods were robust_SKAT, robust_SKATO, p_ttest_asymptopic, p_ssuw_asymptopic, p_ssu_asymptopic. With wider regions (5,000 base pairs), the ranking of the best state-of-the-art methods became robust_SKATO, robust_SKAT, p_ttest_asymptopic, p_ssu_asymptopic and p_wep

As stated above, because of the heterogeneity of the results, these observations are global and on average, and cannot be extended to the two other $\alpha$ levels (S10 Table). These results underscore the inherent difficulty in choosing the right statistical test for a given set of data and analysis, demonstrating once more the usefulness of our ensemble method to guide the user towards the preferred methods.

## Computational time analysis

The computational time of our 4 ensemble methods and 59 state-of-the-art methods depends on the scenario and parameters (see Table 1 and S13 and S12 Figs). For example, the average time for Excalibur on genetic regions of 3 kb and 5 kb in size, is 29 and 52 sec. respectively. The average time to run Excalibur on a cohort of 100 and 5000 individuals is 14 and 31 sec.,

respectively. Based on the minimum and maximum computational times across all 18,000 empirical power simulation, we established the best and worst computational time required to analyze 20,000 genetic regions (the entire exome) for each test (see S13 Table). Excalibur, as a combination of 36 tests, can take up to 458 hours (see S13 Table) to run the entire genome (up to 82 sec. per gene). A standard solution to this limitation is to use parallel computing. For example, using 10 threads of 8 Gb memory each could lower the computational time of such an analysis to less than two days.

## Discussion

In total 59 aggregation tests with each having different underlying assumptions were analysed and compared. We focused on a series of limitations regarding how to prepare data and formulate assumptions when collapsing variants. We showed the impact of different parameter choices on each method's performance. Most importantly, we demonstrated the inability of most tests to control type I error across all scenarios. Results also indicate that it is difficult to describe tests based on their class (Burden, Variance-component and omnibus), because of variability within each class. All the comparative results indicated the difficulty of selecting the right statistical test for a particular set of data and assumptions. The two new tests robust_S-KAT and robust_SKATO performed better than state-of-the-art methods in most experiments.

Based on this issue and our observations, we propose a new ensemble method Excalibur, which incorporates 36 aggregation tests. Unlike most methods, Excalibur was able to control type I error in all simulations. This novel method achieved the best average power across all scenarios that were considered. We showed that the framework is robust and able to overcome the limitations considered in the exhaustive set of comparative simulations. Therefore, Excalibur has a wide range of applicability and is useful to indicate which test would work the best for a particular set of data and assumptions. In WES or WGS analysis, we cannot expect all genetic regions (*e.g.* genes) to follow the same genetic model of association, and therefore, some tests might be more suited for some of the regions, while performing poorly on others. Having an ensemble method that is able to perform such preliminary screening to indicate, which test is the most suitable for which genetic region, is very useful.

The simulations we performed, while being extensive, are not exhaustive. We only considered binary differentiation (cases versus controls), but one could extend this to continuous phenotypes. Some of the tests considered are applicable to quantitative-trait data, while others can only be applied to dichotomous phenotype. One might investigate other covariates as well (*e.g.* gender, age, etc.) and their impact. One could extend the simulations to other genetic regions (*e.g.* pathways). Aside from MAF, predictive bioinformatics tools could offer another source for weights [91–99]. All of these can have an impact of the power and type I error of any of the methods.

Our ensemble methods were constructed based on simulation results of 59 aggregations tests. Exploring a wider or different set of parameters, and including other methods may shift the results, potentially leading to a better ensemble method. One could also investigate more specific ensemble methods. For example, one could build an ensemble method including only aggregation tests dedicated for handling rare variants, and another one focusing on methods dedicated at handling both rare and common variants. The same could apply for dedicated Burden methods and Variance-component methods. We focused on a general ensemble method including a wide range of aggregation tests.

The main limitation of our ensemble method is that it is conservative and computationally intensive. While our primary focus remained fixed upon constructing an ensemble method

that accentuates statistical power, we concede that the avenue of constructing an ensemble method guided by distinct criteria–for example centered around judicious correlation management between tests–warrants further exploration. For example, using the Min(p) approaches that empirically estimates the correlation structure [100]. The trade-off between computational efficiency and statistical power underscores the complex choices that underlie the optimization of ensemble methods.Moreover, we did not explore all the current limitations of aggregation tests, such as incorporating variant annotations to enhance statistical power by assigning functional relevance to variants or addressing the impact of population structure [101]. When using aggregation methods, one should pay attention to various factors (*e.g.* sample selection, coverage harmonization) [102].

In summary, an extensive comparison of aggregation tests has allowed us to propose a new ensemble method based on an optimal combination of 36 tests, leading to the best control of type I error and the best average power across all scenarios. The proposed method will be useful for WES/WGS association studies, as it is able to handle a wide range of association models in order to guide the user towards the optimal aggregation test for a particular genetic region.

## Supporting information

**S1 Fig. Simulation framework.** Schema of the code structure (independent modules represented in blue or green boxes) and data flow (black arrows) of our simulation framework. The green boxes represent steps that are specific to empirical power simulations.
(PNG)

**S2 Fig. Badly Controlled Type I error.** Methods (X axis) that had an inflated type I error for experiment ID n˚4 (Table 1) and their type I error (Y axis) at nominal level $\alpha = 0.05$ based on 10 000 replicates. The red line corresponds to $\alpha = 0.05$ and blue lines correspond to 95% confidence interval. Confidence interval computed assuming that the number of false positives follows a binomial distribution with parameters 10,000 and 0.05. Each bar is colored given the reliability.
(PNG)

**S3 Fig. Well Controlled Type I error.** Methods (X axis) that have a good type I error for experiment ID n˚4 (Table 1) and their type I error (Y axis) at nominal level $\alpha = 0.05$, based on 10 000 replicates. The red line corresponds to $\alpha = 0.05$ and blue lines correspond to 95% confidence interval. Confidence interval computed assuming that the number of false positives follows a binomial distribution with parameters 10,000 and 0.05. Each bar is colored given the reliability.
(PNG)

**S4 Fig. Empirical power of tests with well controlled type I error.** Plots for methods (X axis) having proportion of inflated type I error equal to zero (S4 Table) and their empirical power (Y axis) at nominal level $\alpha = 0.05$ based on 1000 replicates for experiment ID n˚9 (Table 1).
(PNG)

**S5 Fig. Empirical power of tests with badly controlled type I error.** Plots for methods (X axis) with proportion of inflated type I error above zero (S4 Table) and their empirical power (Y axis) at nominal level $\alpha = 0.05$, based on 1000 replicates for experiment ID n˚9 (Table 1).
(PNG)

**S6 Fig. PCA of empirical power of state-of-the-art tests.** Plot of first principal component (X axis) and second principal component (Y axis) of 59 state-of-the-art methods colored by cos2: squared cosine values, indicate the contribution of each variable to a specific principal

component. Higher cos2 values imply a stronger correlation between the variable and the principal component, indicating a better representation of the variable on the plot. The principal component analysis is based on 18,000 empirical power simulations for each test.
(PNG)

**S7 Fig. Heatmap of tests similarities.** Heatmap of similarities, ranging from zero (in blue), to 1 (in red), of our 4 ensemble methods and 59 state-of-the-art methods (X and Y axis). Similarity is defined as the proportion of simulation where two tests give the same output (significant or non-significant) evaluated at nominal level $\alpha = 0.05$ out of the 18,000 empirical power simulations. Only similarities above 0.5 are displayed. Green: test is included in Excalibur. Black: test is not included in Excalibur, or is one of our ensemble methods.
(PNG)

**S8 Fig. Hierarchical clustering of similarities of tests included in Excalibur.** Hierarchical clustering of similarities, ranging from zero (in blue), to 1 (in red), of 36 state-of-the-art methods (X and Y axis) included in Excalibur. Similarity is defined as the proportion of simulation where two test give the same output (significant or non-significant) evaluated at nominal level $\alpha = 0.05$ out of the 18,000 empirical power simulations. Only similarities above 0.5 are displayed.
(PNG)

**S9 Fig. Type I error across cohort sizes.** Plot for our 4 ensemble methods and 59 state-of-the-art methods (X axis) and their type I errors (Y axis) at nominal level $\alpha = 0.05$ for experiment ID n˚1 in red, n˚2 in green, n˚3 in blue and n˚4 in magenta (Table 1). Type I error results based on 10 000 replicates for each experiment. The straight black line corresponds to $\alpha = 0.05$ and dashed black lines correspond to 95% confidence interval. Confidence interval computed assuming that the number of false positives follows a binomial distribution with parameters 10,000 and 0.05. All experiments sare based on exact same parameters except for the cohort size and shows the impact of that parameter on the behavior of all aggregation tests analyzed.
(PNG)

**S10 Fig. Empirical power across cohort sizes.** Plot for our 4 ensemble methods and 59 state-of-the-art methods (X axis) and their empirical power (Y axis) at nominal level $\alpha = 0.05$ for experiment ID n˚1 in red, n˚2 in green, n˚3 in blue and n˚4 in magenta (Table 1). Empirical power results based on 1000 replicates for each experiment. All simulations were based on exact same parameters except for the cohort size and show the impact of that parameter on the behavior of all aggregation tests analyzed.
(PNG)

**S11 Fig. Summary power evolution across scenario.** Plot for our 4 ensemble methods and 59 state-of-the-art methods (X axis) and their total empirical power evolution (Y axis) at nominal level $\alpha = 0.05$ for seven scenarios (see colors), based on S10 Table. Green: evolution of power given the evolution of proportion of case in the cohort, based on empirical power ID n˚14, n˚ 11 and n˚15 (Tables 1 and S5). Black: evolution of power given the evolution of cohort size, based on empirical power ID n˚6, n˚7, n˚8 and n˚9 (Tables 1 and S5). Red: evolution of power given the evolution of proportion of protective variants, based on empirical power ID n˚11, n˚ 12 and n˚13 (Tables 1 and S5). Blue: evolution of power given the evolution of proportion of causal variants, based on empirical power ID n˚2, n˚3, n˚4 and n˚5 (Tables 1 and S5). Magenta: evolution of power given the inclusion of only rare variants versus rare and common variants, based on empiric al power ID n˚18 and n˚10 (Tables 1 and S5). Turquoise: evolution of power given the evolution of causal MAF cutoff, based on empirical power ID n˚1 and n˚11

(Tables 1 and S5). Brown: evolution of power given the evolution of region size, based on empirical power ID n˚17 and n˚16 (Tables 1 and S5).
(PNG)

**S12 Fig. Computational Time Analysis for the Excalibur Method.** Boxplot illustrating the computational time (in seconds) required by the Excalibur method across varying cohort sizes. The x-axis represents the cohort size, while the y-axis denotes the computational time in seconds. The distribution of computational time is visualized using boxplot, providing insights into the method's efficiency as cohort size changes.
(PNG)

**S1 Table. Details methods and packages.** Details of each of the investigated state-of-the-art method, the package and the iteration at which the method was removed or in which version of the ensemble method it was added. 1Hybrid method selects a method based on 3 data points: the total minor allele count (MAC), the number of individuals with minor alleles and the degree of case-control imbalance.
(XLSX)

**S2 Table. Ensemble methods summary.** Details of each state-of-the-art method included in our 4 ensemble methods (in green) and the iteration at which the method was added to the ensemble method. In red, the last iteration that failed to meet the criteria of type I error inflated equal to zero (S4 Table).
(XLSX)

**S3 Table. Results Type I error.** Results of nine type I error experiments evaluated at nominal alpha level of 0.05, 0.01 and 0.001 for our 4 ensemble methods and 59 state-of-the-art methods. When a test failed to return a p-value for an experiment, no information included.
(XLSX)

**S4 Table. Summary Type I error.** Summary table of nine type I error experiments evaluated at three alpha levels (i.e. 0.05, 0.01, 0.001) for our 4 ensemble methods and 59 state-of-the-art methods. 27 type I error results generated per test. Prop exp type I error Good: computed as the number of times a test managed to control type I error divided by 27. Prop exp type I error Inflated: the proportion of time, out of the 27 results, in which a test had an inflated type I error. Prop exp Type I error Conservative: the proportion of time, out of the 27 results, in which a test had a conservative type I error. Prop exp type I error NA: proportion of time a test failed to return a p-value divided by 27. Reliability: defined as the sum of significant and non-significant p-values divided by the number of replicates (i.e. 10000). Reliability min: indicates the lowest reliability among all nine experiments. Reliability max: indicates the highest reliability among all nine experiments. Reliability median: gives the median proportion of time a test returned a p-value divided by the expected returned p-values (i.e. 10 000 per experiment) across all nine experiments.
(XLSX)

**S5 Table. Results empirical power.** Results of 18 empirical power experiments evaluated at nominal alpha level of 0.05, 0.01 and 0.001 for our 4 ensemble methods and 59 state-of-the-art methods. For each of the methods, the minimum, maximum, standard deviation and the average empirical power is display, respectively in columns Min, Max, Sd and Average. The Average empirical power per experiment at the bottom of the table.
(XLSX)

**S6 Table. Ranking empirical power.** Ranking given the power of 18 empirical power experiments evaluated at nominal alpha level of 0.05, 0.01 and 0.001 for our 4 ensemble methods and 59 state-of-the-art methods.
(XLSX)

**S7 Table. Summary ranking.** All our 4 ensemble methods and 59 state-of-the-art methods, ordered by their average ranking. For each empirical power experiment [18] and each alpha level [3], each test is ranked given its power. Based on the 54 rankings, we computed an average, best and worst ranking achieved by each test. Green: methods with proportion of experiments where type I error is inflated is equal to zero.
(XLSX)

**S8 Table. Scenario description.** Description of the seven scenarios, their parameters and the particular values within each experiment ID for both type I error and empirical power. Green: scenario specific to empirical power.
(XLSX)

**S9 Table. Type I error across scenario.** Type I error at nominal level $\alpha = 0.05$, 0.01 and 0.001 for each scenario (see S8 Table) for our 4 ensemble methods and 59 state-of-the-art methods. Evolution of type I error: the difference between two experiments. For example, column 0.2_0.5 is the difference in type I error in between column 0.2 and 0.5 (for scenario 1. Proportion case/control). Column total: the sum of evolution of type I error. Column evolution: the direction of change in type I error.
(XLSX)

**S10 Table. Empirical power across scenario.** Empirical power at nominal level $\alpha = 0.05$, 0.01 and 0.001 foreach scenario (see S8 Table) for our 4 ensemble methods and 59 state-of-the-art methods. Evolution of empirical power: the difference between two experiments. For example, Excalibur (see column "test"), at alpha level 0.05 (see column alpha), for scenario proportion case/control (see column 1. Proportion case/control) has power 0.862, 0.94 and 0.975 when proportion case/control is equal to 0.2, 0.5 and 0.8 respectively. Column 0.2_0.5 is the difference in power in between column 0.2 and 0.5 (for scenario 1. Proportion case/control). Column total_evol: the sum of evolution of empirical power. Column evolution: is the interpretation of column 0.2_0.5 and 0.5_0.8 (*i.e.* the direction of change in power).
(XLSX)

**S11 Table. Cohort size scenario.** Empirical power at nominal level $\alpha = 0.05$ for the cohort size scenario (see S8 Table) for our 4 ensemble methods and 59 state-of-the-art methods. Evolution of empirical power: the difference between two experiments. For example, column 100_200 is the difference in power in between column 200 and 100. Column total: the sum of evolution of empirical power. Column evolution: the direction of change in power.
(XLSX)

**S12 Table. Summary scenario.** Total evolution of empirical power and type I error for our 4 ensemble methods and 59 state-of-the-art methods at nominal level $\alpha = 0.05$ for seven scenarios. Prop case: evolution of power given the evolution of proportion of cases in the cohort, based on empirical power ID n˚14, n˚11 and n˚15 (Tables 1 and S5) and based on type I error ID n˚6, n˚3 and n˚7 (Tables 1 and S3). Cohort size: evolution of power given the evolution of cohort size, based on empirical power ID n˚6, n˚7, n˚8 and n˚9 (Tables 1 and S5) and based on type I error ID n˚1, n˚2, n˚3 and n˚4 (Tables 1 and S3). % protective: evolution of power given the evolution of proportion of protective variants, based on empirical power ID n˚11, n˚12 and n˚13 (Tables 1 and S5). % causal: evolution of power given the evolution of proportion of

causal variants, based on empirical power ID n˚2, n˚3, n˚4 and n˚5 (Tables 1 and S5). Kind of variant: evolution of power given the inclusion of only rare variants versus rare and common variants, based on empirical power ID n˚18 and n˚10 (Tables 1 and S5) and based on type I error ID n˚4 and n˚5 (Tables 1 and S3). causal MAF cutoff: evolution of power given the evolution of causal MAF cutoff, based on empirical power ID n˚1 and n˚11 (Tables 1 and S5). Region size: evolution of power given the evolution of region size, based on empirical power ID n˚17 and n˚16 (Tables 1 and S5) and based on type I error ID n˚8, n˚3 and n˚9 (Tables 1 and S3). For example, Excalibur_baseline (see column Test) has an increase of power of 0.056 when increasing proportion of cases / controls (see column Prop case). This comes from the total_evol column (see S10 Table column total_evol for 1. Proportion case/control for Excalibur_baseline at alpha 0.05).
(XLSX)

**S13 Table. Computational time.** Computational time of our 4 ensemble methods and 59 state-of-the-art tests, based on 18,000 empirical power simulations. The average, minimum and maximum computational time are given in seconds. Best: minimal time needed to perform 20,000 genetic regions based on the minimum computational time, given in hours. Worst: maximal time needed to perform 20,000 genetic regions based on the maximum computational time, given in hours. The laste 18 columns show the average computational time for each empirical power experiment ID (see Table 1), in seconds.
(XLSX)

## Acknowledgments

The authors thank all the members of the laboratory of Human Molecular Genetics and members of the oligogenic team at the Interuniversity Institute of Bioinformatics in Brussels for their support and feedback. We also thank the National Lottery, Belgium and the Foundation against Cancer (2010–101), Belgium for their support to the Genomics Platform of University of Louvain and de Duve Institute, as well as the Fonds de la Recherche Scientifique—FNRS Eguipment Grant U.N035.17 for the «Big data analysis cluster for NGS at UCLouvain». S.B. was supported by fellowships from F.R.I.A. (Fonds pour la formation à la recherche dans l'industrie et dans l'agriculture), and Patrimoine UCL. The authors thank the Genomics Platform of University of Louvain for access to the biocomputing cluster. We also thank the National Lottery, Belgium and the Foundation against Cancer (2010–101), Belgium for their support to the Genomics Platform of University of Louvain and de Duve Institute, as well as the Fonds de la Recherche Scientifique—FNRS Eguipment Grant U.N035.17 for the «Big data analysis cluster for NGS at UCLouvain».

## Author Contributions

**Conceptualization:** Simon Boutry.

**Formal analysis:** Simon Boutry.

**Funding acquisition:** Miikka Vikkula.

**Investigation:** Simon Boutry, Miikka Vikkula.

**Methodology:** Simon Boutry.

**Project administration:** Simon Boutry, Tom Lenaerts, Miikka Vikkula.

**Resources:** Miikka Vikkula.

**Supervision:** Raphaël Helaers, Tom Lenaerts, Miikka Vikkula.

**Writing – original draft:** Simon Boutry.

**Writing – review & editing:** Raphaël Helaers, Tom Lenaerts, Miikka Vikkula.

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
