## [Decision Letter · Decision Letter 0]

5 May 2023

Dear Prof. Vikkula,

Thank you very much for submitting your manuscript "Excalibur: a new ensemble method based on an optimal combination of aggregation tests for rare-variant association testing for sequencing data" for consideration at PLOS Computational Biology.

As with all papers reviewed by the journal, your manuscript was reviewed by members of the editorial board and by several independent reviewers. In light of the reviews (below this email), we would like to invite the resubmission of a significantly-revised version that takes into account the reviewers' comments.

The reviewers acknowledged and appreciated the effort to evaluate several state-of-the-art aggregation tests, but several weaknesses in the evaluation and presentation were brought up by multiple reviewers. The omission of SKAT tests from the comparison, the lack of clarity in how correlation between the different aggregation tests is handled, and lack of details related to computation time for an ensemble method such as Excalibur were cited as major weaknesses. The reviews also suggested a summary of the comparisons that could help a user identify some of the tests suitable for various scenarios would add to the value of the manuscript.

We cannot make any decision about publication until we have seen the revised manuscript and your response to the reviewers' comments. Your revised manuscript is also likely to be sent to reviewers for further evaluation.

Sincerely,

Aakrosh Ratan

Guest Editor

PLOS Computational Biology

Jian Ma

Section Editor

PLOS Computational Biology

The reviewers acknowledged and appreciated the effort to evaluate several state-of-the-art aggregation tests, but several weaknesses in the evaluation and presentation were brought up by multiple reviewers. The omission of SKAT tests from the comparison, the lack of clarity in how correlation between the different aggregation tests is handled, and lack of details related to computation time for an ensemble method such as Excalibur were cited as major weaknesses. The reviews also suggested a summary of the comparisons that could help a user identify some of the tests suitable for various scenarios would add to the value of the manuscript.

Reviewer's Responses to Questions

**Comments to the Authors:**

Reviewer #1: The review is uploaded as an attachment.

Reviewer #2: The authors comprehensively evaluated state-of-art methods and four novel methods performances across different scenarios, and showed the outperformance of Excalibur in controlling type I error while offering the best average power. The authors also investigated each parameter’s effect on the behavior across methods, which provided valuable information for other researchers to choose methods based on different situations in their own research.

One issue that needed be clarified is that the authors claimed that “Excalibur also automates the test selection depending on the genetic region to be explored (e.g. genes, pathways, etc).” However, it is not crystal clear from the construction of the methods on page 13 how Excalibur choose the tests depending on the genes or pathways. Some explanations would be helpful to strengthen this claim.

Also, it would be interesting to see if Exacalibur would also have outstanding performance in the mixed populations with people from different ancestries. The variants effects may differ in size and even in direction in different ancestry populations.

Reviewer #3: The authors try to use an ensemble method, minimum P after multiple test correction, to combine different rare variant tests to achieve a robust and efficient test. However, the comparison design excludes powerful SKAT tests, which is the state-of-the-art methods for rare variant analysis, simply due to strict computation time. The comparison also omits the robust binary SKAT recently developed (Zhao et al., 2020, AJHG). The improved SKAT tests can address the inflated type I error problem when the sample size is small, the allele frequency is too extreme, or the case control balance is too extreme, even though the computation time is increased. This is not a problem because for rare variant test in WES or WGS, we can often scan the whole genomes using a quick version of SKAT and then for interesting genes, we can use a more time consuming resampling version of SKAT to make sure the type I error is controlled. Therefore, I think the comparison is flawed. Here are my suggestions.

1. Include the robust binary SKAT recently developed (Zhao et al., 2020, AJHG) in the comparison.

2. Include those SKAT versions removed due to computing time to have a direct comparison with the proposed ensemble method on type I error, power, and computing time.

3. When comparing power, there is no need to discuss methods that cannot control type I error.

4. Line 473-474: “minimum p-value from the set of tests included in the ensemble method after multiple testing correction using Benjamini-Hochberg”. Is the correction using the Bonferroni correction or Benjamini-Hochberg? The former is commonly used and is robust against correlated test statistics. The latter is used to calculate the false discovery rate (FDR) and in theory it requires independence of test statistics. Whether the FDR adjustment can be used to control type I error in theory is in question.

5. I appreciate that many comparisons have been done. However, it is important to summarize the results that can help users to decide which tests to choose in which scenarios. For example, for each scenario, identify the top 3-5 methods. All methods with obvious inflated type I errors should not be included, which should be removed at the first place.

6. The main figures are all blurred.

**Have the authors made all data and (if applicable) computational code underlying the findings in their manuscript fully available?**

Reviewer #1: Yes

Reviewer #2: Yes

Reviewer #3: Yes

PLOS authors have the option to publish the peer review history of their article (what does this mean?). If published, this will include your full peer review and any attached files.

Reviewer #1: No

Reviewer #2: No

Reviewer #3: **Yes: **Wenan Chen
---

## [Decision Letter · Decision Letter 1]

30 Jul 2023

Dear Prof. Vikkula,

Thank you very much for submitting your manuscript "Excalibur: a new ensemble method based on an optimal combination of aggregation tests for rare-variant association testing for sequencing data" for consideration at PLOS Computational Biology. As with all papers reviewed by the journal, your manuscript was reviewed by members of the editorial board and by several independent reviewers. The reviewers appreciated the attention to an important topic. Based on the reviews, we are likely to accept this manuscript for publication, providing that you modify the manuscript according to the review recommendations.

Sincerely,

Aakrosh Ratan

Guest Editor

PLOS Computational Biology

Jian Ma

Section Editor

PLOS Computational Biology

Reviewer's Responses to Questions

**Comments to the Authors:**

Reviewer #1: The review is uploaded as an attachment.

Reviewer #2: The authors revised the manuscript nicely, and explained investigating Excalibur performance on mixed populations is out-of-scope, which is acceptable.

Some comments on line 290: suggest to use "log" instead of "ln" to make it consistent with line 288 where it uses log(MAF). Also suggest to delete "0,402" since the rest cs calculated with different MAFs do not show an associated approximate value. Correct the format for the first c.

Reviewer #3: Here are my future concerns which are important to assess the quality of the manuscript.

1. Table 1 is not complete in the PDF file, so it is hard to check in which scenario which methods are performing well and whether the results make sense.

2. The powers of the proposed methods are much higher than the previous version. The previous proposed methods all have power < 0.6. The suggested Robust binary SKAT is already > 0.6 in the new version. Is it due to the inclusion of the suggested robust binary SKAT recently developed (Zhao et al., 2020, AJHG)? Please add comments or discussions in the paper.

3. The type I error of the robust binary SKAT/SKATO is 0 when the nominal value is 0.05. This is different from the paper (Zhao et al., 2020, AJHG) which shows that the type I error is very close to the nominal level. Can the authors explain? Is it because the sample size or other settings are different? It will be more convincing if the authors can show that when the simulation setting parameters are changed to that used in (Zhao et al., 2020, AJHG), then the type I error is close to the nominal.

4. Include those SKAT versions removed due to computing time to have a direct comparison with the proposed ensemble method on type I error, power, and computing time. Here is the list to consider because these methods is likely to be more powerful even though more time is needed.

pBin_linear_SKATO_ER

pBin_linear_weighted_SKATO_ER

pBin_linear_SKATO_ERA

pBin_linear_weighted_SKATO_ERA

pBin_linear_SKATO_Hybrid

pBin_linear_weighted_SKATO_Hybrid

5. Besides the proposed methods, it will add more value to the paper to summarize the results to identify the top 3-5 individual methods for each scenario because users may only consider running one individual method instead of an ensemble method.

**Have the authors made all data and (if applicable) computational code underlying the findings in their manuscript fully available?**

Reviewer #1: Yes

Reviewer #2: Yes

Reviewer #3: Yes

PLOS authors have the option to publish the peer review history of their article (what does this mean?). If published, this will include your full peer review and any attached files.

Reviewer #1: No

Reviewer #2: No

Reviewer #3: No

Figure Files:

Data Requirements:

Reproducibility:

References:

---

## [Editor Report · Decision Letter 2]

4 Sep 2023

Dear Prof. Vikkula,

We are pleased to inform you that your manuscript 'Excalibur: a new ensemble method based on an optimal combination of aggregation tests for rare-variant association testing for sequencing data' has been provisionally accepted for publication in PLOS Computational Biology.

Best regards,

Aakrosh Ratan

Guest Editor

PLOS Computational Biology

Jian Ma

Section Editor

PLOS Computational Biology

---

## [Editor Report · Acceptance letter]

11 Sep 2023

PCOMPBIOL-D-23-00145R2 

Excalibur: a new ensemble method based on an optimal combination of aggregation tests for rare-variant association testing for sequencing data

Dear Dr Vikkula,

I am pleased to inform you that your manuscript has been formally accepted for publication in PLOS Computational Biology. Your manuscript is now with our production department and you will be notified of the publication date in due course.

With kind regards,

Jazmin Toth
